# Effect of Organic Polymers on Mechanical Property and Toughening Mechanism of Slag Geopolymer Matrix

**DOI:** 10.3390/polym14194214

**Published:** 2022-10-08

**Authors:** Xiaotong Xing, Jiangxiong Wei, Weiting Xu, Beihan Wang, Shunjie Luo, Qijun Yu

**Affiliations:** School of Materials Science and Engineering, Guangzhou 510641, China

**Keywords:** geopolymer, SCAs, water-soluble polymers, porosity, mechanical property, polymerization mechanism

## Abstract

In this work, two series of chemically reactive polymers, silane coupling agents (SCAs) and water-soluble polymers, were specifically designed as an additive to improve the ductility of slag geopolymer paste by vibration pressure technique. The influences of organic polymers on the fluidity, rheological behavior, mechanical property, porosity, and toughening mechanism of slag geopolymer were investigated. The polycondensation and bonding characteristics of organic–inorganic products were calculated by ^1^H liquid nuclear magnetic resonance (NMR) technology and Fourier transform infrared (FT-IR). The polymerization degree of composite geopolymer was evaluated by ^29^Si NMR and X-ray photoelectron spectroscopy (XPS). The microscopic morphology of the geopolymer matrix was analyzed using scanning electron microscopy (SEM). The results showed that the dosage of the KH570 and PAA-Na with 5 wt% behaved best in improving the flexural strength and the compressive strength of geopolymer in their corresponding organic series, respectively. The addition of polymers decreased the fluidity and the fluidity loss ratio of geopolymer slurry but reduced the harmful pores of hardened geopolymer. The organic polymers acting as bridge-fixed water molecules weakened the repulsion force, and formed a three-dimensional network through molecular interweaving in a geopolymer matrix. Methacryloxy in silane coupling agents and carboxyl group in water-soluble polymers may contribute to the improvement of hydration product structure through strong bonding with C-A-S-H. Microscopic measurements indicated that the addition of KH570 and PAA-Na in geopolymer could form 73.55% and 72.48% Si-O-Si with C-A-S-H gel, higher than the reference, and increase the polycondensation degree of C-A-S-H phase, reflected by the increased generation of Q^2^ and Q^2^(1Al) and the longer chain length, leading to a higher densified geopolymer matrix with high ductility.

## 1. Introduction

Inorganic cementitious materials have been widely used in various fields of civil engineering. However, traditional inorganic cementitious materials are typical brittle materials, with low flexural strength (1/20–1/10 of the compressive strength) and poor toughness because of the internal chemical bonds including ionic bonds, covalent bonds, and van der Waals force which have poor resistance to plastic deformation [1,2,3]. Due to the low ductility, they are prone to cracking under low tensile strains, typically between 10^−4^ and 10^−3^ [1]. The cracks weaken the structural-bearing capacity and overall stability and limit the application scope of cementitious materials [4,5]. With the rapid development of the social economy and complexity of the building structure, growing demand for high-toughness inorganic cementitious materials has been raised. The addition of a fiber-modified method via the microcrack bridging effect is currently the effective method for increasing the tensile and flexural strengths and toughening concrete through physically blocking crack propagation [6]. However, the incompatibility between fiber and geopolymer, incorporation and homogenization of fiber in an inorganic matrix limit the application, and the incorporation of fibers does not change the brittle fracture nature of cementitious composites.

Inspired by natural bio-composites, such as nacre, which is composed of 99 vol% hard and brittle aragonite (CaCO_3_) layers separated by 1 vol% soft protein layers, which are 1000 times as tough as its constituent CaCO_3_ phase, the strategy of introducing soft substances into the hydration system to hybridize the hydrates was proposed to improve the strength and toughness of inorganic cementitious materials. The introduction of organic molecules or polymers into the inorganic matrix has drawn considerable attention in recent years [7,8,9]. Polymers-modified inorganic materials can fill the pore defects, bridge the fine cracks, and generate an interpenetrating network with cement hydrates, hence improving the toughness [10].

In 1980, Birchall [11,12] and Oludele [13] demonstrated that the incorporation of polymer in cement can achieve 60-70MPa flexural strength by mitigating macroscopic flaws during preparation, using traditional polymer processing by rolling, extrusion and dough mixing with a low water/cement ratio of 0.1 to 0.2. Birchall proposed the definition of “macroscopic defect-free (MDF) cement” for this organic–inorganic composite cement. [11]. The calcium aluminate cement–polyvinyl alcohol polymer (CAC-PVA) system was one of the examples with very high flexural strength up to 150 MPa, close to that of ordinary steel [14,15,16]. The cross-linking structure formed between organic polymer and inorganic matrix was proved to be the reason for the high flexural strength of MDF cement. The high pH environment of hydrating cement causes the acetate group to quickly hydrolyze until PVAC is completely hydrolyzed. The acetate ions subsequently react with the calcium ions to form calcium acetate. In addition, the aluminum ions react with the hydrolyzed PVA to form a cross-linked network. For ordinary Portland cement-based materials, Feng et al. [17] studied the effects of different types of silane coupling agents (SCA) and their derivatives on the hydration process and mechanical properties of cement paste. They found that the epoxy SCA improved the compressive strength and splitting tensile strength of cement paste by 20% and 38%, respectively. Based on previous research, the introduction of organic polymers improved the low-fractures strength and brittle behavior of cement-based materials. In recent years, the toughening issues of geopolymer, which is a green cementitious material, have gained much attention. Zhang et al. found the incorporation of polyvinyl alcohol and polyethylene into metakaolin-based geopolymers achieved noticeably ameliorated mechanical performance [18]. The addition of polyacrylic acid (PAA) and sodium polyacrylate (PAA-Na) can improve the compressive strength by 25.4 MPa (maximum: 29.0%) and cross-bending strength by 6.1MPa (maximum: 64.9%). Chen et al. [19] compared polyethylene glycol (PEG), polyacrylamide (PAM) and sodium polyacrylate (PAAS) to the mechanical properties of slag-based geopolymer, and they found that the bending toughness coefficient of the geopolymer concrete incorporating 0.6wt% PAAS was increased by 53.7%. Wang et al. [20] revealed the toughening modification mechanism of fly ash-based geopolymers by epoxy resins, which play a role in the filling and crossing-linking based on XRD and SEM analyses.

Calcium aluminosilicate hydrate (C-A-S-H) gel is the major phase in ordinary geopolymer hydration products and is responsible for cohesion and mechanical properties [1,21]. Based on the experimental and theoretical studies, the nanostructure of C-A-S-H is composited of tetrahedral anions [SiO_4_]^4−^ and [AlO_4_]^5−^ sharing bridging oxygen (known as “Dreierketten”-like silicate tetrahedra chains) and forms a layered structure resembling a tobermorite analog. The interlayer region is filled with alkali cations such as (Ca^2+^ and K^+^) compensating for the electric charge and water molecules [22,23]. It is apparent that the C-A-S-H gel structurally accommodates a substantial concentration of defects, such as the absence of tetrahedrons in the interlayer. Under the action of alkali activator, the geopolymer precursor undergoes depolymerization and condensation processes, which provides conditions for forming a cross-linked network with organic polymers. The key to improving the mechanical performance is the polymeric bonding between different functional groups of organic polymers and major hydration products at the molecule level.

Polymers with polar chemical groups can react with cementitious materials in aqueous solutions. Water soluble polymers with carboxyl groups, amino groups or hydroxide radicals were proved to be conducive to improving the mechanical properties of the cementitious matrix [18]. SCAs coupling agents (SCAs) with both organic and inorganic attachments can chemically bond with inorganic materials under alkaline aqueous conditions. The typical structure of SCAs is (RO)_3_-Si-R′-X, where the “X” group represents organic functional groups, e.g., alkyl, epoxy and vinyl, etc., “R′” represents a small alkylene linkage, such as –CH_2_CH_2_ and “RO-” represents a hydrolyzable group such as an alkoxy group. “OR” groups form covalent bonds with C-A-S-H to increase the flexural strength of the cementitious matrix [14,24,25]. 

However, few researchers took slag as a starting material to prepare geopolymer products toughening organic polymers. In addition, the investigations on the content of chemical adsorption and the types and the proportions of different chemical reactions by the addition of different series organic polymers with different functional groups were not systematical and were just limited to the qualitative analysis level. Therefore, in this paper, the polymerization mechanism of organic polymers with C-A-S-H gel in slag geopolymer was investigated. Three types of SCAs containing different polar functional groups (aminopropyl-based SCA named KH550, epoxy-based SCA named KH560 and methactyloxy-based SCA named KH570) and two types of water-soluble polymers including polyvinyl alcohol (PVA) with hydroxyl groups and sodium polyacrylate (PAA-Na) with carboxyl groups were selected to be used as the ductility-enhancing agent for geopolymer. Their organic molecular structures are shown in Figure 1. The mechanical properties and the porosity of slag geopolymer incorporated with organic polymers were determined. The physical adsorption capacity and the chemical reactivity of polymers with C-A-S-H were calculated according to ^1^H liquid NMR testing results. The organic–inorganic polymerization degree of composite products was evaluated by FTIR and ^29^Si NMR tests. The effect of the different organic functional groups on the bonding characterization of final complex products was determined by XPS analysis. The morphologies of the composite geopolymer samples were analyzed by scanning electron microscope (SEM). 

## 2. Materials and Methods

### 2.1. Materials Used

The chemical composition of the slag used throughout the experiment is given in Table 1. The alkali activator was water glass (industrial sodium silicate) provided by Foshan Zhongfa Company, China. The modulus of the water glass was 3.2, and the solid content was 37.92%. The water glass contained 8.94% Na_2_O and 28.93% SiO_2_. NaOH used to adjust the modulus of water glass was of analytical purity and manufactured by Tianjin Jinhui Taiya Chemical Reagent Co., Ltd. Reagent grades Na_2_SiO_3_·5H_2_O, Ca(NO_3_)_2_·4H_2_O and Al(NO_3_)_3_·9H_2_O were used to synthesize C-A-S-H gel and purchased from Shanghai Macklin Chemical Reagent Co. Ltd. They were dissolved in deionized water at a concentration of 0.1 mol/L, 0.08 mol/L and 0.02 mol/L, respectively, in order to obtain C-A-S-H gel with C/S = 0.8 and A/S = 0.2, which is consistent with the composition of the C-A-S-H gels produced by the alkali activation reaction of the real slag-based geopolymer.

The 3-aminopropyl triethoxysilane (KH550), 3-glycidyloxypropyl trimethoxy-silane (KH560) and 3-methacryloxypropyl trimethoxysilane (KH570) were provided by Shanghai Macklin Chemical Reagent Co., Ltd., Shanghai, China. The water-soluble polymer PVA1788 was a chemical pure agent and was provided by Shanghai Chenqi Chemical Technology Co., Ltd., Shanghai China. The water-soluble polymer PAA-Na (AR) was provided by Tianjin Komiou Chemical Reagent Co., Ltd., Tianjin, China.

### 2.2. Preparation and Mechanical Property of Organic–Inorganic Composite Geopolymer Paste

Before the mechanical testing, slag geopolymer paste with and without organic polymers was prepared by a vibration pressure machine at 25 °C according to C109/C109M [26]. The mixture proportions of slag geopolymer paste incorporating different organic polymers are listed in Table 2. In the experiment, the mass of slag was 4000 g in each group and the mass ratio of water to slag was 0.25. Reference was designed as the control group with no polymer content. Reference1–Reference 4 were mainly used to investigate the effects of the modulus ratio of alkali activator on the mechanical properties of slag-based geopolymer. The concentration of alkali activator remained constant at 6 wt%, respectively, and the modulus ratio of alkali activator was adjusted to 0.8, 1.2, 1.6 and 2.0, respectively. Reference 5–Reference 8 were used to investigate the effects of the concentration of alkali activator on the mechanical properties of slag-based geopolymer. The concentration of the alkali activator was adjusted from 2, 4, 6 to 8 wt% at a constant modulus ratio of the alkali activator according to the strength. The polymer was added at dosages of 1, 3, 5 and 7 wt% of slag to produce the series of geopolymer compositions. The organic polymers were dissolved or mixed into about 30 wt% of water and stirred for 20 min, and BaCl_2_ as the set retarder was dissolved by the remaining water and stirred for 20 min before adding into the paste. The alkali activator solution, the organic polymer emulsion and the BaCl_2_ solution were added to the slag and gradually stirred for 1 min with low velocity until a homogeneous slurry was formed, and then the mixture was stirred for 1 min with high velocity. Afterward, the mixture was cast into a 300 × 300 × 25 mm iron mold of the vibration pressure machine shown in Figure 2. The specific height was determined according to the filler content. To mitigate large pores of paste specimens, the mixtures were compacted for 5 min with a vibration pressure of 18H_Z_. Afterward, the casted specimens were cured by steam curing at 80 °C for 4 h and then cured at 25 °C temperature for an additional 7 and 27 days [27,28]. The test block of each mixing proportion was cut into 5 pieces with sizes of 250 × 50 × 25 mm by a slicing machine. The compressive strength and flexural strength results were determined by the average of 5 blocks.

### 2.3. Sample Preparation of Organic–Inorganic Composite Geopolymer Mixture for Polymerization Reaction Degree Study

The physical adsorption and chemical bonding characteristics of organic polymers to slag geopolymer was investigated by ^1^H liquid NMR testing. The organic–inorganic composite geopolymer mixtures were prepared with a high water/solid ratio of 2.5. The alkali content of the activator was calculated as Na_2_O equivalent. The dosage of organic polymers was according to the result of the strength. The mixture samples were pre-treated and eluted in 25 °C water for 4 h and 100 °C water for 4 h to exclude the physically absorbed organic polymers.

### 2.4. Sample Preparation of Organic–Inorganic Composite Geopolymer for the Polymerization Mechanism Study

To explore the polymerization mechanism between the organic polymers and C-A-S-H in slag at the microscopic level, C-A-S-H gel was synthesized with the incorporation of different kinds of organic polymers through the direct precipitation method. The Ca/Si ratio and Al/Si ratio was the typical proportioning dose from the literature [29]. The preparation process of polymer/C-A-S-H was the same as that of the C-A-S-H gel. The polymer concentration was 9.44 g/L. The preparation scheme of polymer/C-A-S-H composite samples is illustrated in Figure 3. The nitrate solution was added into sodium silicate solution dropwise to achieve the desired starting composition. NaOH particles were added to adjust the pH of the synthetic solution near 13 PH. The mixed solution was stirred at 90 rpm for 24 h. During this process, nitrogen was constantly injected into the solution to prevent carbonization. The solids were subjected to three alternating cycles of centrifugation and elution with deionized water. The filtered samples were then dried in a vacuum with a desiccator for 7 days to a constant mass. After that, the filtered samples were ground in an agate mortar and sieved with a 75-micron sieve for the subsequent analyses of FTIR, ^29^Si NMR, XPS and field emission SEM. 

### 2.5. Experimental Methods

#### 2.5.1. Fluidity, Mechanical and Porosity of Organic–Inorganic Composite Geopolymer

The fluidity of the organic–inorganic composite geopolymer slurry was determined in accordance with the Chinese Standard GB/T 8077-2012 [30]. The flow diameter of geopolymer slurries was measured by a flow cone with size of 60 mm height, 36 mm top diameter and 60 mm bottom diameter. After the cone was taken out vertically from the slurry sample, the maximum diameter and the maximum width vertically perpendicular to the diameter of the diffusion sample were measured. The average value of these two values was defined as the fluidity value. The fluidity loss ratio was calculated in Equation (1):Fluidity loss ratio % = (F_0_ − F_30min_)/F_0_%(1)

The compressive strength and flexural strength of hardened paste were determined by the fully automatic bending compression testing machine (UTM5105). The maximum force value was 100 kN. The loading speeds for the compressive strength and flexural strength test were 2 mm/min and 0.3 mm/min, respectively.

The porosity and pore size distribution of hardened paste were determined by mercury intrusion porosimeter (MIP). The paste samples were crushed to small pieces about 1 cm^3^ and dried in a vacuum oven at 50 °C to constant weight.

#### 2.5.2. Polymerization Reaction Degree Test

The polymerization reaction degree between different organic functional groups and hydration products of geopolymer was quantitatively identified by the liquid-state ^1^H nuclear magnetic resonance (NMR, AVANCE III HD 600, Bruker, Zurich, Switzerland) spectra analysis. The testing frequency was 600 MHz. The deuterated dimethyl sulfoxide (C_2_D_6_SO) was used as a solvent, and the acetonitrile (C_2_H_3_N) was used as an internal standard substance. The organic–inorganic composite geopolymer mixture solution was prepared with mixed proportions as described in Section 2.3. The mechanical, physical and chemical adsorption amounts of organic functional groups on C-A-S-H gel were calculated by the internal standard absolute determination method. The mass percentage of the polymer was quantified by comparing the integral of the peaks for protons of different kinds of organic polymers and acetonitrile [31]. In the first stage, the mixed solution of organic matter and C-A-S-H gel was left to stand for 7 days, and then the eluted supernatant was subjected to the liquid NMR test to calculate the mechanical adsorption amounts. In the second stage, the precipitate which was filtrated in the first stage was rinsed in water at 25 °C for 4 h, and then the eluted supernatant was sent for liquid NMR test to calculate the physical adsorption amounts. In the third stage, the precipitate which was filtrated in the second stage was rinsed in water at 100 °C for 4 h, and then the eluted supernatant was determined by liquid NMR to calculate the chemical adsorption amounts.

#### 2.5.3. Polymerization and Toughing Mechanism Test

The polymerization and toughing mechanism of solid polymer/C-A-S-H gel were evaluated, which helps to establish the links between microstructure and mechanical properties [32]. The type of chemical bond and functional group of organic–inorganic composite geopolymer paste was detected by Fourier transform infrared spectrometer (FTIR, Nicolet IS50-Nicolet Continuum, Thermo Fisher Scientific). The FTIR tests were conducted in a transmission model in which the scan resolution was 4 cm^−1^, and each measurement was recorded based on 32 scans. The real Ca/Si ratio and Al/Si ratio of paste were calculated through the fusion method based on the elemental analysis results determined by an X-ray fluorescence spectrometer (XRF, Axios PW4400, PANalytical B.V, Almelo, Netherlands). The mass ratio of the paste sample to the solvent (lithium tetraborate) was 1:14. The paste sample and solvent were fused to the glass plate for testing at 1050 °C. The polymerization reaction degree of polymer/C-A-S-H gel was tested by ^29^Si MAS NMR spectra via Varian INOVA-300 spectrometer (7.05 T). The ^29^Si NMR spectrum was scanned 5000 times with a rotation rate of 5 kHz, a relaxation time of 5 s and single pulse width of 6 µs. In addition, the PeakFit 4.0 software was used to perform peak fitting processing of experimental data on the original map. To accurately separate the peak results, the AutoFit Peaks III Deconvolution program was used, and the relevant parameter R^2^ was greater than 0.99. In this research, the morphology of the organic–inorganic interpenetrating network was observed by FE-SEM (SU8220, HITACHI). It was not sprayed with gold to reflect the original morphology of the sample. The working distance was 12 mm, and the voltage was 10 kV under the secondary electron (SE) mode. The elements, bonds and chemical states analysis of composite geopolymer pastes were monitored by X-ray photoelectron spectroscopy (XPS, Axis Ultra DLD, Kratos Analytical Ltd., Chest-nut Ridge, NY, Manchester, UK) with monochromatic Al Kα (hν = 1486.7 eV) X-ray source operating at 150 W (10 mA, 15 kV) and were charged corrected to the main line of the carbon C 1s spectral component (C-C and C-H bonds) set to 284.80 eV. The analysis area of spectrums was 300 × 700 µm. The Kratos charge neutralizer system was used throughout. The high-resolution spectrums were measured with the step size of 0.1 eV and 20 eV pass energy. Instrument base pressure was 2 × 10^−8^ Pa. A standard Shirley background was used for all sample spectrums, which can calculate the chemical reaction ratio between organic polymer and C-A-S-H gel. 

## 3. Results and Discussion

### 3.1. Mechanical Properties of Organic–Inorganic Composite Geopolymer

#### 3.1.1. Effects of Alkali Concentration and Modulus Ratio of the Alkali Activator on the Mechanical Properties of Slag-Based Geopolymer

The toughening effect of organic polymers on slag-based geopolymer was examined by the mechanical properties of the hardened paste. The molded sheets were cut into the sizes of 25 × 25 × 25 mm and 250 × 50 × 25 mm for compressive strength and flexural strength, respectively. The modulus ratio of the alkali activator had a remarkable effect on the mechanical properties of the geopolymer. Figure 4 shows the strength of slag-based geopolymer with the different modulus ratios. It can be seen that the compressive and flexural strengths of slag-based geopolymer both increase first and then decrease when the modulus ratio of the alkali activator increases from 0.8 to 2.0. The mechanical properties reach the maximum values when the modulus ratio is 1.6. At this modulus ratio, the compressive and flexural strengths of the specimens after aging for 28 d are, respectively, 70.34 MPa and 5.39 MPa. In the case of lower modulus (SiO_2_/Na_2_O) ratios of alkali activator, the increase in modulus ratio leads to more Si ions in the system, which can enhance the degree of polymerization and generate a denser silicon-rich gel phase [32]. While the modulus ratio increased, the content of Na_2_O decreased and the dissolution of silicon–aluminum raw materials was less enhanced [33].

In addition, the concentration of alkali activator also plays an important role in affecting the mechanical properties of geopolymer. The strength of geopolymer always varies greatly with different concentrations of alkali activator [34,35]. As Figure 5 showed, it can be seen that with the increase of the concentration from 2 wt% to 8 wt%, the compressive and flexural strengths of slag-based geopolymer increased and then decreased; the majority of compressive and flexural strength happened when alkali content was 6%. The result is consistent with Sukmak et al. [36] and Leong et al. [37]. 

#### 3.1.2. Effect of Polymer Content on the Mechanical Properties of Organic–Inorganic Composite Geopolymer

The compressive strength and the flexural strength of organic-inorganic composite geopolymers at the age of 7 and 28 d are shown in Table 3. The 7 d and 28 d compressive strengths of slag geopolymer without organic polymer are 63 MPa and 70.34 MPa, respectively. The 7 d and 28 d flexural strengths are only 4.21 MPa and 5.39 MPa, respectively. It can be noted that with the increase of different chemically reactive polymers content from 1 to 7 wt%, the flexural and compressive strengths increased steadily and then decreased. When the content of organic polymers is 5 wt%, the compressive and flexural strength is higher than the reference specimens and other polymer content. The compressive strength of KH570/slag geopolymer increased to 124.99 MPa and 129.31 MPa at the curing age of 7 d and 28 d, respectively. It is observed that the flexural strength of KH570/slag geopolymer at 7 d and 28 d is 8.56 MPa and 10.88 MPa, respectively, higher than those of KH550/slag geopolymer and KH560/slag geopolymer. In addition, the 7 d and 28 d compressive strength of PAA-Na/slag geopolymer is 100.67 MPa and 109.34 MPa, respectively. The 7 d and 28 d flexural strengths are 7.24 MPa and 8.31 MPa, respectively, higher than those of PVA/slag geopolymer. It indicates that the incorporation of polymers, especially PAA-Na and KH570, has a positive effect on the flexural and compressive strengths. However, when the polymer content was more than 5 wt%, the mechanical properties of slag-based geopolymer began to degrade. The reason for this strength trend may be due to the excessive polymers covering the surface of unreacted slag and hydration products, which hindered further hydration and limited cross-linking between hydration products. In addition, higher polymer content prompted an increase in system viscosity and pore size. Thus, the dosage of 5% different chemically reactive polymers was used for subsequent experiments.

### 3.2. Fluidity of Organic–Inorganic Composite Geopolymer Slurry

The effect of incorporation of different chemically reactive polymers on the workability of slag geopolymer slurries and the fluidity and fluidity loss ratio were determined, and the results are shown in Figure 6a,b. The fluidity was measured during the first 30 min due to the sensitivity of the water content of the vibration pressure machine. Compared with the reference, the fluidity of geopolymer slurry decreased by 17.5%, 19.32% and 20%, respectively, with the addition of 5% KH550, 5% KH560 and 5% KH570, while the addition of PVA and PAA-Na did not decrease the fluidity results, just by 5.5% and 4.5%. It is shown that the incorporation of the organic polymers makes the time-dependent loss ratio decrease, especially in the case of SCAs/geopolymer slurry, to 17.31% with 5% PVA, 16,67% with 5% PAA-Na, 6.33% with 5% KH550, 3.94% with 5% KH560 and 6.25% with 5% KH570. The main reason for the reduction of fluidity and time-dependent loss ratio may be attributed to the fact that under alkaline conditions, the polarized groups of chemically reactive polymers such as hydroxyl, carboxyl groups, and high-activity silanol can form a hydrogen bond with the O^2-^ and adsorption layer with Ca^2+^ on the surface of the slag particles. In addition, HO- groups of polymers can also react with C-A-S-H gel. The formed organic–inorganic bonding layers inhibit hydration evolution and delay coagulation of C-A-S-H gel to some extent at an early age. 

### 3.3. Porosity and Pore Size Distribution of Organic-Inorganic Composite Geopolymer

The porosity and pore size distribution of organic–inorganic composite geopolymer paste were shown in Figure 7a–d. The pore volume distribution of paste is shown in Figure 7e,f. The pore diameter is divided into four categories [38]: harmless pores (d < 20 nm), little harmful pores (d = 20–50 nm), harmful pores (d = 50–200 nm), and much harmful pores (d > 200 nm). As shown in Figure 7a,c, the organic–inorganic composite geopolymer pastes have lower capillary porosity in comparison to the reference paste. The cumulative pore size distributions are given in Figure 8b,d. It is seen that the cumulative pore volume of the reference is about 0.035 mL/g. The total pore volume of the incorporation of chemical reactive polymers is nearly 0.015 mL/g and 42.86% of the reference. As shown in Figure 7e,f, the total pore volume of PVA/geopolymer, PAA-Na/geopolymer, KH550/geopolymer, KH560/geopolymer and KH570/geopolymer are 28.58%, 42.85%, 99.99%, 38.58% and 42.38% of the reference. The proportion of harmful pores volume larger than 50 nm of PVA, PAA-Na, KH550, KH560 and KH570 are 40.58%, 24.28%, 99.8%, 20.58% and 38.33% of the reference. The same conclusion can be drawn from Figure 7b,d. The addition of chemically reactive polymers can reduce the overall pore size and total porosity of the geopolymer matrix due to the formation of an organic–inorganic cross-linking network structure.

### 3.4. The Polymerization Degree between Different Functional Groups of Polymers and Hydration Products

The polymerization degree between different functional groups of polymers and hydration products of geopolymer was investigated by the liquid-state ^1^H NMR experiment [39], and the results are shown in Figure 8 and Figure 9. The mechanical, physical and chemical adsorption content of SCAs in C-A-S-H gel of slag geopolymer is shown in Figure 8a,d,g, where the black spectrum represents pure SCAs, and the red spectrum represents the supernatant of KH550/slag geopolymer solution, KH560/slag geopolymer solution and KH570/slag geopolymer solution. Resonance at the chemical shift of about 1.38 ppm represents CH_2_-CH_2_-CH_2_ of KH550. Resonance at approximately 3.64 ppm is attributed to CH_2_-CH_2_-CH_2_ of KH560, and resonance at 6.03 ppm and 5.67 ppm stands for CH_2_=C(CH_3_)COOR of KH570. According to Figure 8a,d,g, all red spectra do not show characteristic peaks of SCA, indicating the supernatant-free SCAs were all absorbed in the hydration products. 

After removing the supernatant, the precipitates were subjected to water washing treatment at 25 °C and 100 °C, the content of SCAs in solution as shown in Figure 8b,c,e,f,h,i. According to the nuclear magnetic MestReNova integral, the content of the SCAs in the elution solution was quantified by comparing the peak intensity of the proton in acetonitrile according to the Formulas (2) and (3).
(2)$$W_{s}={A*W}_{a}*\frac{m_{s}}{m_{a}}$$
(3)A=ISnsIana
where *I_s_, I_a_* is the integrated intensity, and *n_s_, n_a_* is the amount of hydrogen containing the absorption peak of each molecule.

The calculation results are shown in Table 4. It is seen that all of KH570 is proved to be chemically absorbed by the hydration products. The mechanical adsorption amount of KH550 is 0.19%, and the physical adsorption amount is 0.11% by weight. In contrast, the mechanical adsorption amount of KH560 is 18.58%, and the physical adsorption amount is 10.42% by weight. The variation of the functional group of amino-based, epoxy-based and methacrylate-based SCAs is responsible for the adsorption content difference of polymers on C-A-S-H. The KH570 has fewer mechanical and physical adsorption content, but larger chemical adsorption content than KH550 and KH560, suggesting that KH570 is better at participating in the formation of organic–inorganic chemical bonding and network structure.

The mechanical, physical and chemical adsorption content of water-soluble organic polymers (PVA with a hydroxyl group and PAA-Na with carboxyl) in C-A-S-H gel of slag geopolymer was determined by the liquid-state ^1^H NMR experiment. The testing results are shown in Figure 9a,d, where the black spectrum represents pure polymer, and the red spectrum represents the supernatant. The characteristic resonance peak range was from 3.35 ppm to 3.98 ppm and from 3.28 ppm to 3.38 ppm, respectively. According to the spectrum and calculation, both polymers are present in the supernatant, and the concentration is 9.4% and 5.87% by weight, respectively. It is worth noting that no characteristic resonance peak is found in the elution solution with PVA and PAA-Na after washing at 25 °C and 100 °C as shown in Figure 9b,c,e,f. For PVA with OH- and PAA-Na with COO-, after washing treatment, the remaining water-soluble polymer was chemically adsorbed by the hydration products of geopolymer. Based on the testing results from the perspective of quantitative analysis, KH570 with methacrylate and PAA-Na with carboxyl group with a relatively high chemical reaction capability is proposed to be used as a toughening admixture in slag composites. The specific types of chemical reactions are detailed in Section 3.5.

### 3.5. Polymerization Mechanism of Organic Polymers and Slag Geopolymer

#### 3.5.1. XRF Analysis

The synthesis of organic–inorganic composite powders with strictly controlled chemical composition was used to study the organic and the inorganic interaction mechanism. The polymer concentration was 9.44 g/L. The oxide composition of each composite powder formulation was determined by XRF as presented in Table 5. 

#### 3.5.2. FT-IR Spectral Analysis

The chemical bonding characteristic between the organic polymer and hydration products analysis was characterized by FT-IR. The spectra of composites are shown in Figure 10. The band at around 1650 cm^−1^ and 3450 cm^−1^ corresponds to the bending and stretching vibrations of H-O-H, respectively, owing to the presence of water [40]. The Si-O-Al symmetrical stretching vibration band is at approximately 1440 cm^−1^. The intensity of the peak decreases with the incorporation of organic polymers, indicating the chemical ambiance around the Al atom is affected by the organic–inorganic polymerization reactions [41]. As shown in Figure 10a, the spectrum of the C-A-S-H/KH550 at about 1502 cm^−1^ is observed because of the bending vibration of N-H. The strong band at 1000 cm^−1^ is shifted to higher wavenumbers, indicating the formation trend of the C-N band. The spectrum of the C-A-S-H/KH560 at 773 cm^−1^ presents the (CH_2_)_3_ in Figure 11b. The C-A-S-H/KH570 shows more extra peaks at about 1720 cm^−1^, 1371 cm^−1^ and 1298 cm^−1^ which are caused by the C=O stretching band. The Si-CH_2_ asymmetrical stretching vibration and the C-O stretching band are shown in Figure 10c,d. The change of the -OH stretching bond at about 3440 cm^−1^ shifted to a lower wavenumber, demonstrating the formation trend of new hydrogen bonds. The presence of these additional characteristic peaks demonstrates the organic polymer is chemically adsorbed in the C-A-S-H of slag geopolymer. As shown in Figure 10a–c, the main peak at 992 cm^−1^ represents the asymmetric stretching vibration of Si-O-Si (Al). It is the main structural unit of the aluminosilicate network. The intensity of the bond Si-O-Si (Al) increases with the incorporation of SCAs. The reason is that the highly reactive silanol in SCAs polymerizes with Si-O groups in aluminosilicate.

The C-A-S-H/PVA shows more extra peaks at around 2978 cm^−1^, 2864 cm^−1^, 1548 cm^−1^,1446 cm^−1^, 1136 cm^−1^, 831cm^−1^, which are associated with the CH_3_, CH_2_, C-O, C=C-OR, Si-O-C and CH, respectively in Figure 11a. The asymmetric vibration of C-O in COO^-^ indicates that acetate carboxylate precipitates after PVA hydrolysis, and the metal ions in the hydration product form acetate [41]. The generation of chemical bonds Si-O-C demonstrates that polymer changes the chemical environment of Si-O bond. The hydroxy in PVA can form a bond with the non-bridging oxygen in the aluminosilicate sheet. The network structure is constructed between organic polymer and geopolymer. Compared with the asymmetric stretching vibration of the Si-O-Si (Al) peak at 992 cm^−1^ in the reference sample, the corresponding peak of the C-A-S-H/PVA shifts to the left at 982 cm^−1^ [18], suggesting an increase in polymerization degree of aluminosilicates in geopolymer.

The characteristic vibration bands of PAA-Na and C-A-S-H/PAA-Na are observed at 2856 cm^−1^ (C=C) and 1640 cm^−1^ (C=O). For the spectrum at 1424 cm^−1^ and 1418 cm^−1^, since the stretching vibrations of (O=C-OR) and CO_3_^−2^ appear in a very close region, these two bands overlap and become one peak in the spectra of C-A-S-H/PAA-Na materials shown in Figure 11b. FTIR spectra of C-A-S-H and C-A-S-H/PAA-Na materials contained characteristic bands in the range from 992 cm^−1^ to 1008 cm^−1^. These intensive bands in all spectra are assigned to the Si-O stretching vibration of Q^2^ tetrahedra. The wavenumber of the top peak shifts rightward which may be due to the generation of the Si-O-C bond, suggesting the formation of an organic–inorganic interpenetrating network structure. It also indicates the modification of slag-based geopolymer by different chemical reactive groups occurs at the molecular level, which can enhance the flexural toughness of nature.

#### 3.5.3. XPS Spectral Analysis

The chemical state of characteristic elements (Si, C, O and Ca) associated with participation in organic–inorganic polymerization of composite geopolymer was determined by XPS as shown in Figure 12, Figure 13, Figure 14, Figure 15, Figure 16 and Figure 17. As seen in Figure 12a, the reference C-A-S-H shows a prominent peak of Si 2p at 102.3 ± 0.8 eV [41,42]. The presence of Si-OH has a high binding energy compared with Si-O-Si [43,44]. The reaction degree between SCAs and C-A-S-H was detected by the Si (2p) spectrum in XPS shown in Figure 13a, Figure 14a, Figure 15a. With the increase of the proportion of bridging silicon (O-Si-O), the atoms’ electrons in the vicinity of existing -O-Si-O bonds that shield the silicon nuclei result in a detectable chemical shift. It is seen that the bridging silicon proportion of C-A-S-H/KH570 in Figure 15a is 75.18% greater than that of the reference geopolymer (51.85%) in Figure 12a, C-A-S-H/KH550 (63.82%) in Figure 13a and C-A-S-H/KH560 (47.32%) in Figure 14a. The result indicates that KH570 behaves better than KH550 and KH560 in strengthening the polymerization degree of C-A-S-H. The O1s XPS deconvolution spectra of reference C-A-S-H, C-A-S-H/KH550, C-A-S-H/KH560 and C-A-S-H/KH570 are shown in Figure 12b, Figure 13b, Figure 14b and Figure 15b, respectively. The incorporation of KH550 and KH570 leads to more formation of Si-O-Si bridging oxygen in the C-A-S-H matrix. The proportion of bridging oxygen in C-A-S-H/KH550 and C-A-S-H/KH570 is 56.18% and 73.55%, respectively and higher than that of the reference C-A-S-H (51.85%) in Figure 12b. This confirms the increase of bridge oxygen of C-A-S-H, which is consistent with the trend of Si 2p testing results. The increasing amount of bridge oxygen suggests a more complex network structure. However, the KH560 seems not conducive to the increase of C-A-S-H polymerization degree, reflected by the lesser proportion of Si-O-Si bridging oxygen in C-A-S-H/KH560 than that of the reference C-A-S-H. It means that the hydroxyl polycondensation capacity between KH560 with epoxy-based and C-A-S-H gel is weaker. The result is in good agreement with the results determined by ^1^H NMR testing results. 

As shown in Figure 15c, the existence of KH570 is proved by the characteristic peaks of CH_2_=C(COO^−^)CH_3_ at 285.12 eV and COO^−^ at 286.08 eV. In addition, the KH550 and KH560 have the characteristic peak of NH_2_-CH_2_^−^ at 285.13 eV and the epoxy group at 285.31 eV in Figure 13c and Figure 14c, respectively. The detected characteristic chemical environment of C1s is in line with the testing results of FTIR.

The XPS deconvolution spectra of the water-soluble polymer incorporated C-A-S-H are shown in Figure 16 and Figure 17. The O1s and C1s spectrum of C-A-S-H/PVA are shown in Figure 17. The peak at 532.42 eV belongs to the bridging oxygen O^0^, and the area is 64.11% in Figure 16a and higher than that of the reference C-A-S-H (54.76%) in Figure 12b. The results indicate that PVA fills the vacancy of the tetrahedral position by forming more Si-O-C with C-A-S-H. Further demonstration is provided by the C1s spectrum shown in Figure 16b. The C1s spectrum of C-A-S-H/PVA shows elemental carbon with different chemical environments, where 46.59% peak area belongs to the C-OH groups (285.39 eV) and 4.37% peak area is attributed to the non-hydrolyzed COO^-^ groups (288.86 eV). The small amount of detectable COO^-^ groups may be attributed to the original contained raw synthetic materials of PVA. The C1s spectrum results suggest that a polymerization reaction takes place between PVA and C-A-S-H.

The interaction between PAA-Na and hydration products of geopolymer was explained by the acetate and the inorganic ions produced, which are confirmed in the Ca 2p spectrum with the component (Ca(COO)_2_) at 348.5 eV, compared in Figure 17a,c [44]. In addition, in the C1s spectrum in Figure 17c, the precipitation of COO^-^ at approximately 288.8 eV indicates the formation of Ca (COO)_2_, indicating the oxygen atoms in the sodium polyacrylate exhibit affinity with surrounding Ca^2+^ in C-A-S-H gel. Due to the formation of the Ca-O bond, the Ca^2+^ can link PAA-Na to different chains of C-A-S-H. Compared with the O1s spectra of C-A-S-H/PVA as shown in Figure 16a, C-A-S-H/PAA-Na has a higher amount of bridging oxygen O^0^ at around 532.25 eV as shown in Figure 17b, which demonstrates that organic functional groups can react with the non-bridging oxygen in free [SiO_4_]^4−^ tetrahedra and affect the chemical environment of O1s, leading to the formation of a more complex network structure and higher polymerization degree of the C-A-S-H matrix.

Consistent with previous research [45,46], the typical structure of SCAs molecules has three “RO” groups to form three “OH” hydroxyl groups through a hydrolysis reaction. The bridging connection of the organic polymer and inorganic geopolymer can be established by covalent bonds formed between the OH- groups of the SCAs and the OH- groups on the surface of C-A-S-H. In addition, the SCAs can also link C-A-S-H between different layers. Methacrylic-based KH570 provides the best chemical coupling between C-A-S-H gels to enhance cohesion. For the water-soluble polymer, the percentage of Si-O-Si is 72.48% in PAA-Na which is higher than that of PVA (64.11%). Furthermore, the formation of Ca (COO)_2_ between PAA-Na and Ca^2+^ of C-A-S-H gel further enhances the integration of the organic–inorganic network structure.

#### 3.5.4. ^29^Si NMR Spectral Analysis

The location and structural role of the various organic species are characterized by ^29^Si MAS NMR analysis. Figure 18 presents the ^29^Si NMR spectrum of C-A-S-H/ polymer, which was deconvolved into serval peaks. The deconvolution results and relative quantification of C-A-S-H/ polymer are summarized in Table 6. The existing state of different chain lengths of aluminosilicates, where Q^1^, Q^2^, Q^3^ and Q^4^ represent the chain end group, the middle group, the branching site and the fully cross-linked group, respectively, can be revealed. The chemical environment and the polymerization degree of [SiO_4_]^4−^ tetrahedra in geopolymers are affected by incorporating different organic functional groups. Compared with the reference C-A-S-H, polymer/C-A-S-Hs as shown in Figure 18a–f have a sharp Q^2^ peak, and there is a significant increase in the Q^2^/Q^1^ as seen in Table 6. These results suggest that the aluminosilicate chain is extended, and the polymerization degree of aluminosilicate in geopolymer is increased.

The experimental results are in line with the previous research [36], and the mechanism of increased aluminosilicate polymerization is confirmed, that SCAs or water-soluble polymers may be grafted onto the missing sites of [SiO_4_]^4-^ tetrahedra. It is also possible that some polymers intercalate into the interlayer space of C-A-S-H [46]. The variations of the chemical shift of different polymers may be attributed to the chemical attachments such as OH- or COOH-. In the case of PVA and PAA-Na, the mean length of the aluminosilicate chain is 12.14 and 7.78 determined by the high Q^2^/Q^1^ ratio [47]. According to XPS, the -COOR in PAA-Na and Ca^2+^ in the slag are bonded together to form Ca(COO)_2_ so that the main chain length does not increase significantly. The chemical shift of Si in the vicinity of three types of SCAs with different “X” groups proves that C-A-S-H/SCAs obtain more Q^2^ Si-O-Si bonds. The Q^2^/Q^1^ratio of C-A-S-H/KH550, C-A-S-H/KH560 and C-A-S-H/KH570 are 1.97, 1.31 and 2.37, respectively. It is evident that SCAs are grafted onto the aluminosilicate, and KH570/C-A-S-H behaves best in bonding the missing sites of bridging silica tetrahedra along the aluminosilicate chains.
Mean chain length (MCLa )=I(Q1(0Al)+I(Q2(0Al)+32I(Q2(1Al)) 12I(Q1(0Al))
$$\mathrm{Al}/\mathrm{Si}(C\text{-}A\text{-}S\text{-}H)=\frac{Q^{3}\left(1Al\right)}{Q^{1}+Q^{1}\left(1Al\right)+Q^{2}+Q^{2}\left(1Al\right)+Q^{3}+Q^{3}\left(1Al\right)}$$

#### 3.5.5. Morphology Analyses

Secondary electron (SE) images of the synthesized C-A-S-H/polymers samples dried for 7 days are shown in Figure 19. It is seen that the reference C-A-S-H gel in Figure 19a exhibits a globular morphology. The basic structural unit of C-A-S-H is the nano-scale particles that could gather into clusters by a chemical bond. From Figure 19b, it is apparent that there is no connection between the spherical gel particles in the reference C-A-S-H gel. However, particle aggregation is observed in KH550/C-A-S-H and KH560/C-A-S-H samples as shown in Figure 19c,d. For KH570/C-A-S-H in Figure 19e,f, larger size agglomeration appears, showing a clear tendency toward forming an interpenetrating network among C-A-S-H particles. The C-A-S-H/PVA and the C-A-S-H/PAA-Na samples are presented in Figure 19g,h, the nano-sized particles show cross-linking between spherical particles compared with the reference sample, indicating that the bonding effect occurs between water-soluble polymers and C-A-S-H, revealing the intrinsic cause of the improvement of the toughness of organic–inorganic composites. 

## 4. Conclusions

The effects of SCAs and water-soluble polymers on fluidity, the mechanical property and porosity of slag geopolymer were evaluated. The polymerization degree, polymerization mechanism and toughening mechanism between organic polymers (SCAs and water-soluble polymers) and hydration products of slag geopolymer were investigated. Based on the results and discussion, the following conclusions are drawn.

(1)The addition of organic polymers in slag geopolymer showed general enhancement in compressive strength and flexural strength after the curing age of 7 d and 28 d. With the increase of polymer content from 1 to 7 wt%, the compressive strength and the flexural strength were increased first and then decreased. When the polymer was 5 wt%, the optimal mechanical properties appeared in the slag/KH570 paste with compressive strength of 129.31 MPa and the flexural strength of 10.88 MPa at 28 d. The incorporation of silane coupling agents (SCAs) and water-soluble polymers led to a general reduction of pore diameter and pore volume of geopolymer matrix which could improve its compactness. The addition of SCAs led to an obvious decrease in the fluidity of geopolymer slurry, while the incorporation of water-soluble polymers did not change the fluidity results. The fluidity loss ratio decreased with the addition of polymers, especially in the case of SCAs/geopolymer slurry. The organic polymers fixed most of the free water molecules and converted them into gel water which filled the pores between the slag particles to enhance the cohesion of the slurry. In addition, different chemical reactive groups provided more connection sites for reaction with C-A-S-H, forming a three-dimensional network.(2)The polymerization degree between different functional groups of polymers and hydration products was investigated from the quantitative analysis perspective. The chemical adsorption capacity of different organic polymers to hydration products of slag geopolymer by the liquid-state ^1^H NMR spectra was investigated. In the case of SCAs, KH570 exhibited the best chemical adsorption capacity with almost complete adsorption, followed by KH550 with 99.70% and KH560 with 71.00%. For the water-soluble polymers, PAA-Na presented better adsorption capacity, with 94.13% to slag geopolymer, than PVA with 90.60%. Therefore, methacryloxy in silane coupling agents and carboxyl group in water-soluble polymers may be attributed to the improvement of the C-A-S-H structure through the strong bond with the hydration product.(3)The specific chemical reaction toughening mechanism of polymers-reinforced slag-based geopolymer was investigated at the molecular and microscopic levels. The change of chemical environment around the [SiO_4_]^4−^ tetrahedra was determined by ^29^Si NMR. Comparing the KH570/C-A-S-H with the reference C-A-S-H, a sharper Q^2^ peak and a longer mean chain length appearing at KH570/C-A-S-H indicate the aluminosilicate chain was longer because of the increasing degree of polymerization. The O-Si-O bridging silicon proportion of KH570/C-A-S-H with 75.18% was higher than that of the KH550/C-A-S-H (68.88%), the KH560/C-A-S-H (63.82%) and the reference C-A-S-H (51.85%). KH570 with methacrylic behaved best in bonding C-A-S-H gel particles in the series of silane coupling agents forming a three-dimensional network with a higher polymerization degree.(4)The spectra of C-A-S-H modified with the water-soluble polymer showed vibrational peaks of the C-O-Si bond at around 1136 cm^−1^, indicating that the condensation reaction occurred between the -OH functional group in PVA and the -OH in C-A-S-H. The formation of the Si-O-Si (Al) bond and the participation in the construction of the organic–inorganic interpenetration network structure were the main reason for the improvement of the flexural strength and toughness of slag geopolymer. PAA-Na/C-A-S-H and PVA/C-A-S-H showed a C-O-Si bond with 64.11% and 72.48% bridging oxygen, respectively. In addition, the acetate ions of PAA-Na reacted with the calcium ions of C-A-S-H to form 17.24% calcium acetate. PAA-Na with carboxyl group exhibited a better outcome in increasing the toughness of slag geopolymer as compared to PVA. In summary, the experimental results showed that PAA-Na with the carboxyl group had a higher polymerization reaction degree with C-A-S-H gel in the series of water-soluble polymers.(5)The research focuses on the qualitative and quantitative characterization of the chemical reaction processes between organics with different functional groups and geopolymers in order to establish a theoretical foundation for subsequent organic-toughening geopolymers. In the series of silane coupling agents (SCAs), the “X” group and the “RO” groups of silanes with a double bond and “−CH_3_” were selected to bridge the inorganic components which can promote the toughness of slag-based geopolymer. Additionally, in the series of water-soluble polymers, the “COO-” group was selected to improve the toughness of slag-based geopolymer.

## Figures and Tables

**Figure 1 polymers-14-04214-f001:**
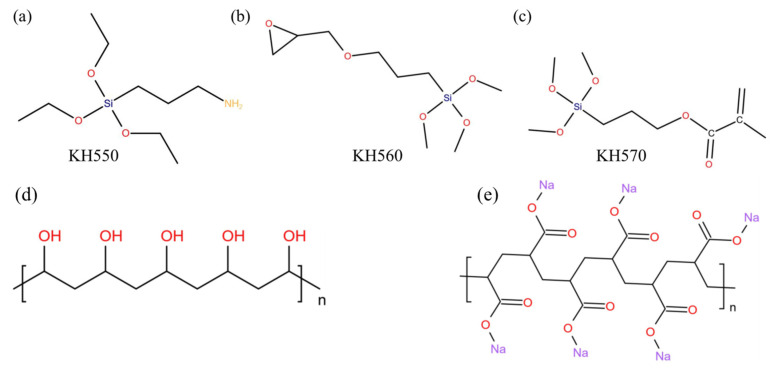
Molecular structure of (**a**) KH550; (**b**) KH560; (**c**) KH570; (**d**) PVA; (**e**) PAA-Na.

**Figure 2 polymers-14-04214-f002:**
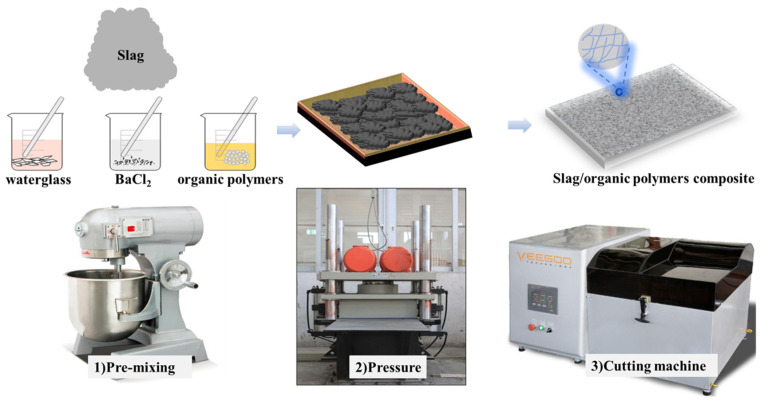
Schematic illustration of fabrication process of organic–inorganic composite geopolymer paste.

**Figure 3 polymers-14-04214-f003:**
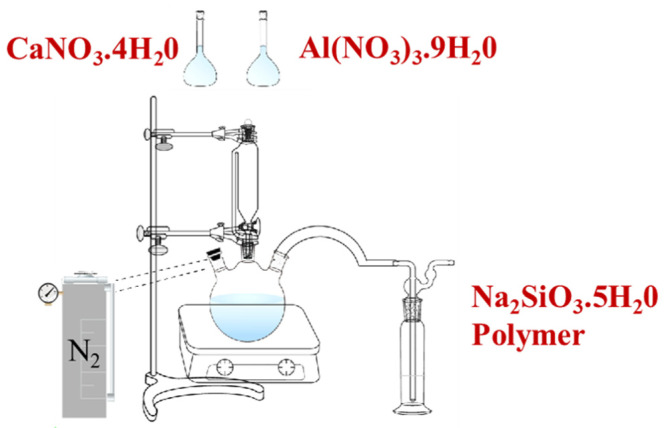
The preparation scheme of polymer/C-A-S-H composite samples.

**Figure 4 polymers-14-04214-f004:**
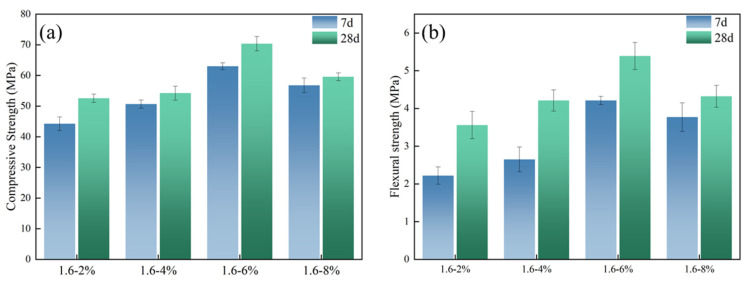
Effect of alkali content on mechanical properties of slag-based geopolymer: (**a**) Compressive strength; (**b**) Flexural strength.

**Figure 5 polymers-14-04214-f005:**
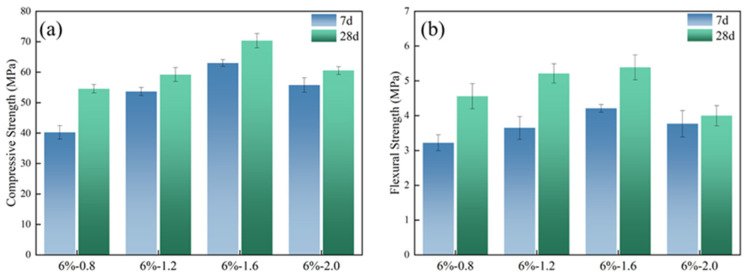
Effect of modulus on mechanical properties of slag-based geopolymer: (**a**) Compressive strength; (**b**) Flexural strength.

**Figure 6 polymers-14-04214-f006:**
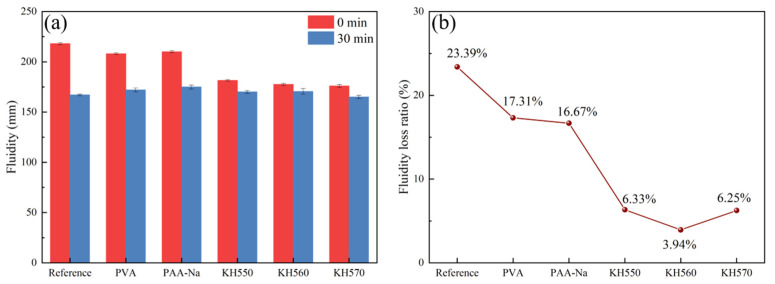
The fluidity (**a**) and fluidity loss ratio (**b**) of organic–inorganic composite geopolymer slurry.

**Figure 7 polymers-14-04214-f007:**
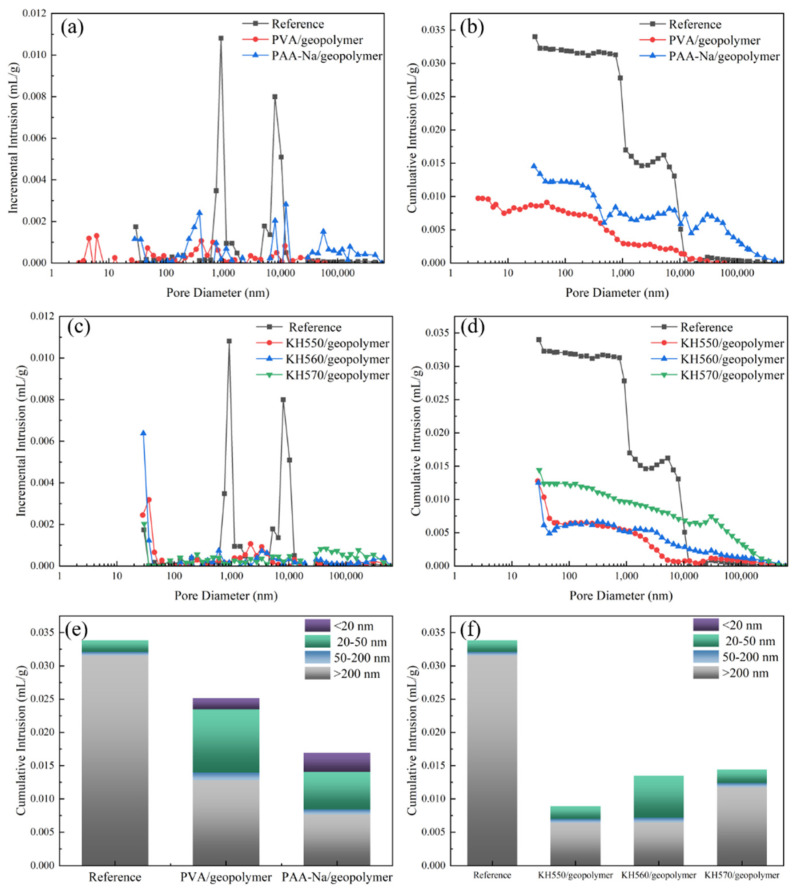
Porosity (**a**,**c**), pore size distribution (**b**,**d**) and pore volume distribution (**e**,**f**) of organic–inorganic composite geopolymer slurry.

**Figure 8 polymers-14-04214-f008:**
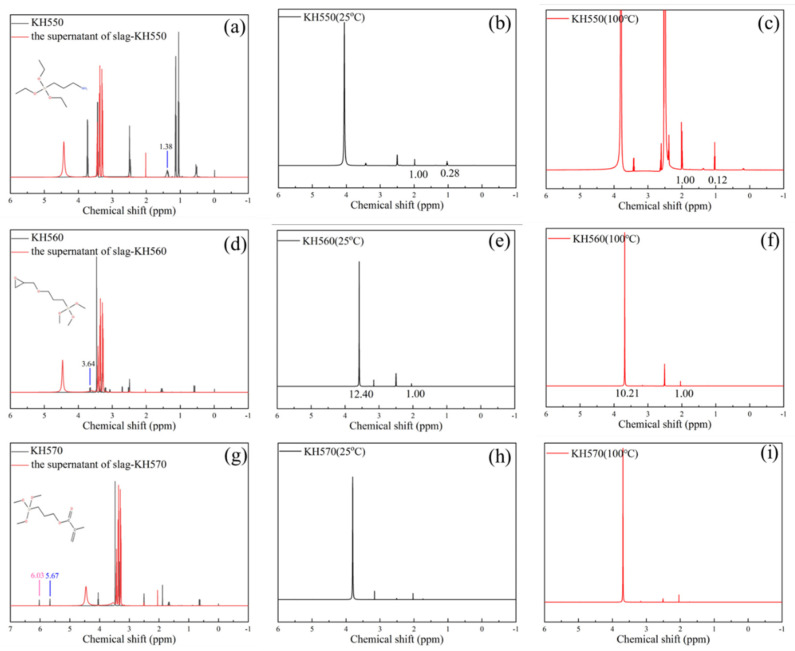
(**a**–**i**) Liquid-state ^1^H NMR spectrum of (**a**) KH550; (**b**) KH550 (25 °C); (**c**) KH550 (100 °C); (**d**) KH560; (**e**) KH560 (25 °C); (**f**) KH560 (100 °C); (**g**). PVA; (**h**) KH570 (25 °C); (**i**) KH570 (100 °C).

**Figure 9 polymers-14-04214-f009:**
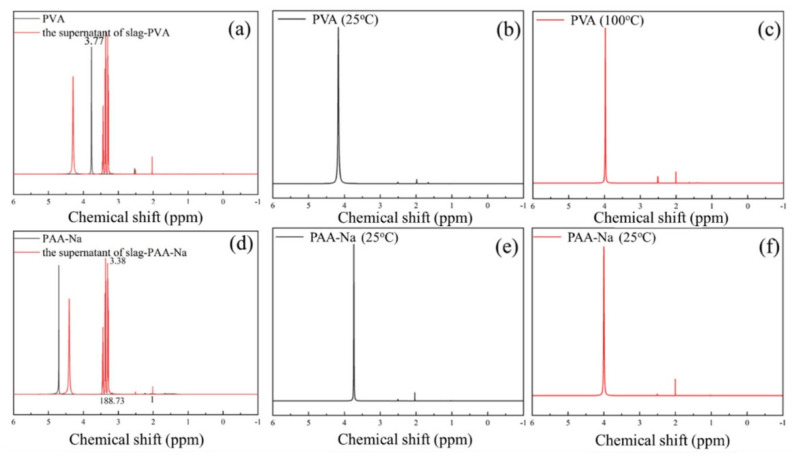
Liquid-state ^1^H NMR spectrum of (**a**) PVA; (**b**) PVA (25 °C); (**c**) PVA (100 °C); (**d**) PAA-Na; (**e**) PAA-Na (25 °C); (**f**) PAA-Na (100 °C).

**Figure 10 polymers-14-04214-f010:**
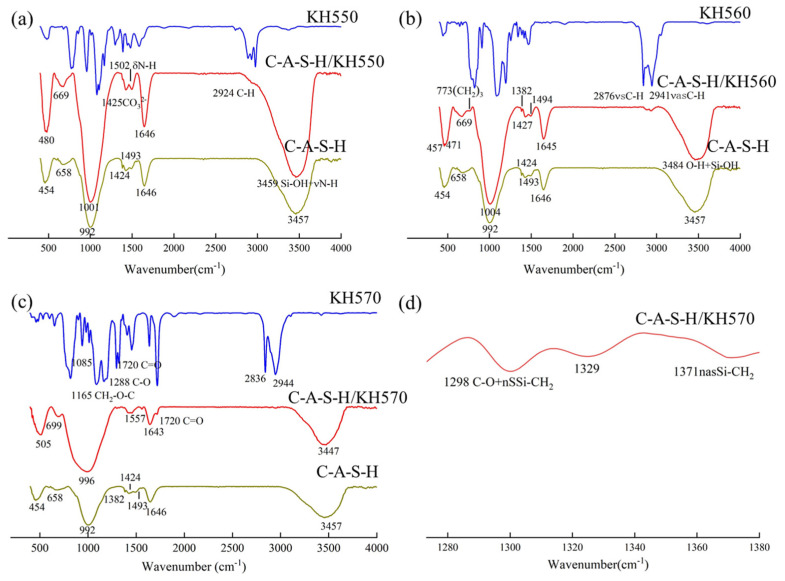
FTIR spectra of (**a**) C-A-S-H/KH550; (**b**) C-A-S-H/KH560; (**c**) C-A-S-H/KH570; (**d**) C-A-S-H/KH570 (Local amplification spectrum).

**Figure 11 polymers-14-04214-f011:**
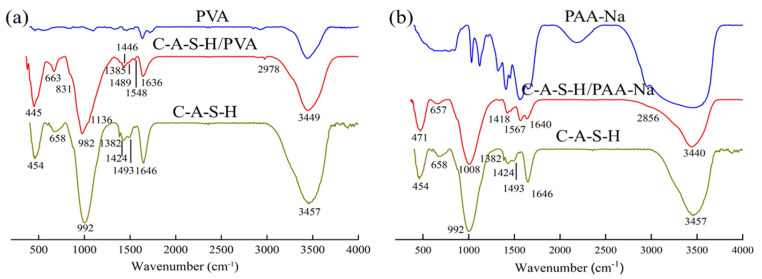
FTIR spectra of (**a**) C-A-S-H/PVA; (**b**) C-A-S-H/PAA-Na.

**Figure 12 polymers-14-04214-f012:**
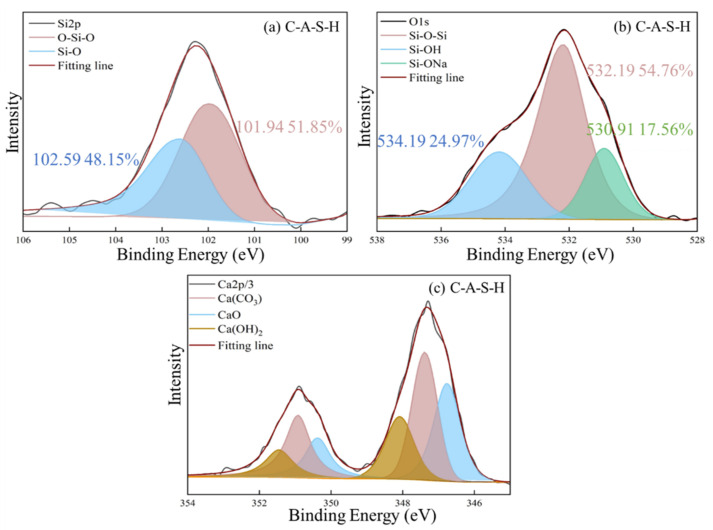
The XPS deconvolution spectrum of (**a**) Si 2p, (**b**) O1s, (**c**) Ca 2p/3 of reference C-A-S-H.

**Figure 13 polymers-14-04214-f013:**
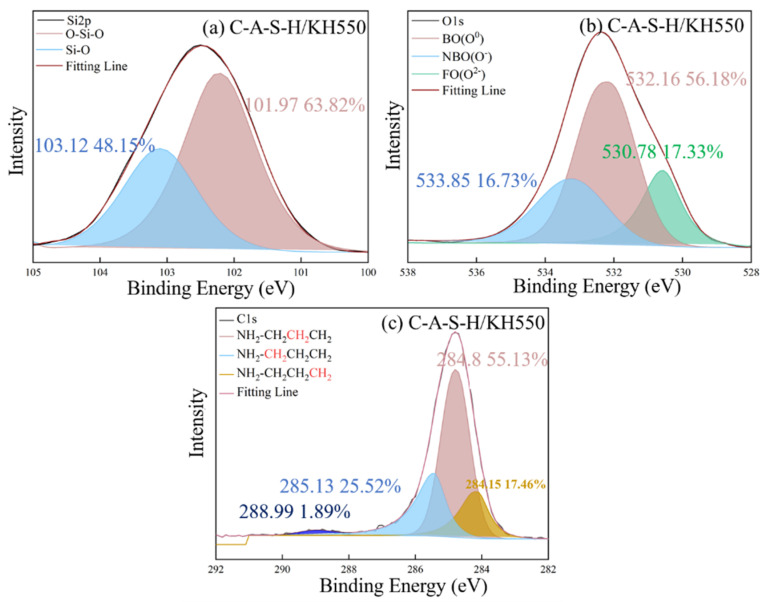
The XPS deconvolution spectrum of (**a**) Si 2p, (**b**) O1s XPS, (**c**) Ca 2p/3 of C-A-S-H/KH550.

**Figure 14 polymers-14-04214-f014:**
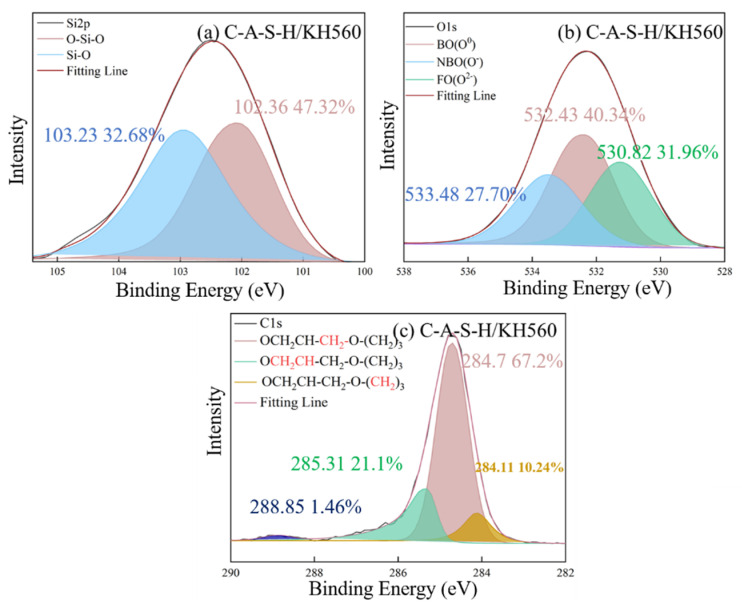
The XPS deconvolution spectrum of (**a**) Si 2p, (**b**) O1s, (**c**) Ca 2p/3 of C-A-S-H/KH560.

**Figure 15 polymers-14-04214-f015:**
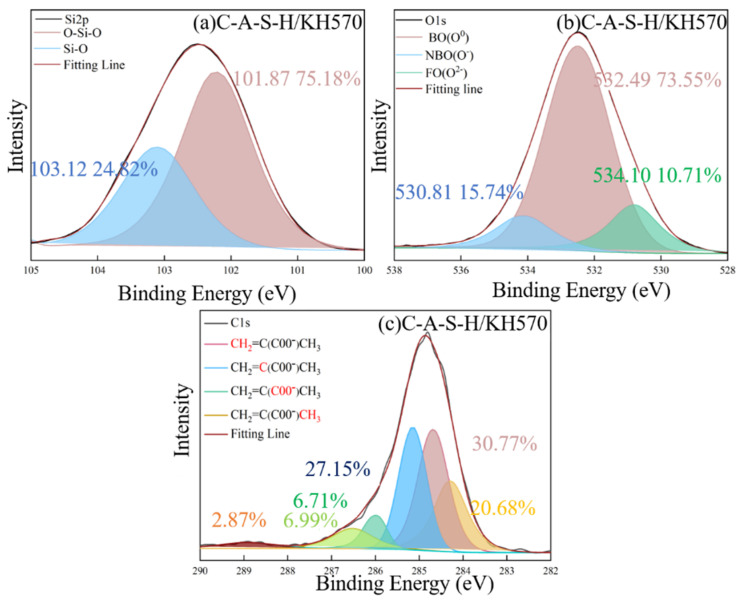
The XPS deconvolution spectrum of (**a**) Si 2p, (**b**) O1s, (**c**) Ca 2p/3 of C-A-S-H/KH570.

**Figure 16 polymers-14-04214-f016:**
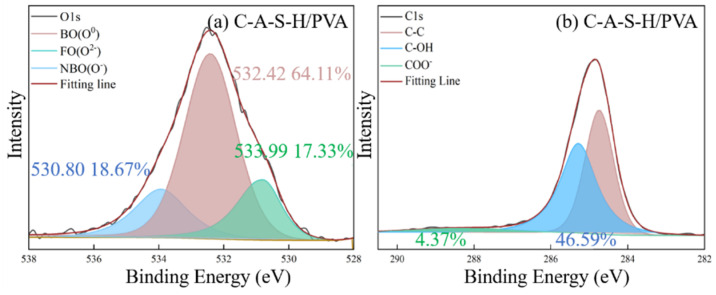
(**a**) The XPS deconvolution spectrum of the O1s and (**b**) C1s of C-A-S-H/PVA.

**Figure 17 polymers-14-04214-f017:**
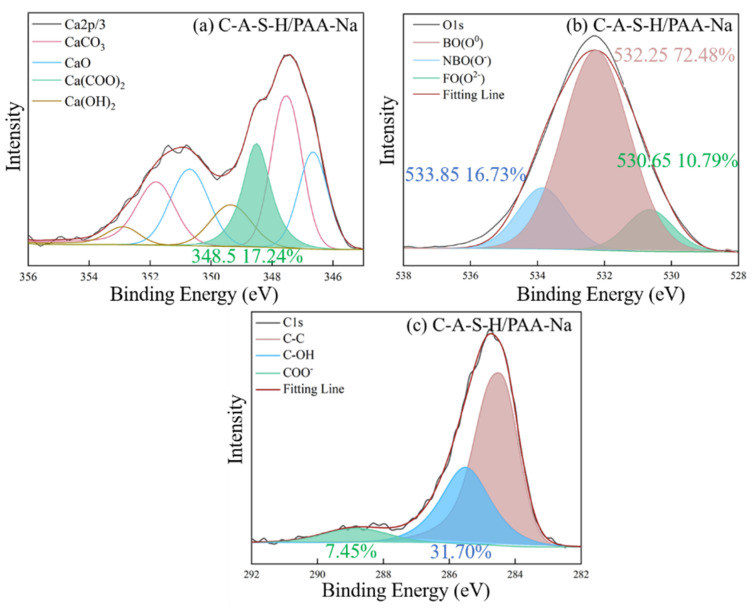
The XPS deconvolution spectrum of (**a**) Ca 2p/3, (**b**) O1s and (**c**) C1s of C-A-S-H/PAA-Na.

**Figure 18 polymers-14-04214-f018:**
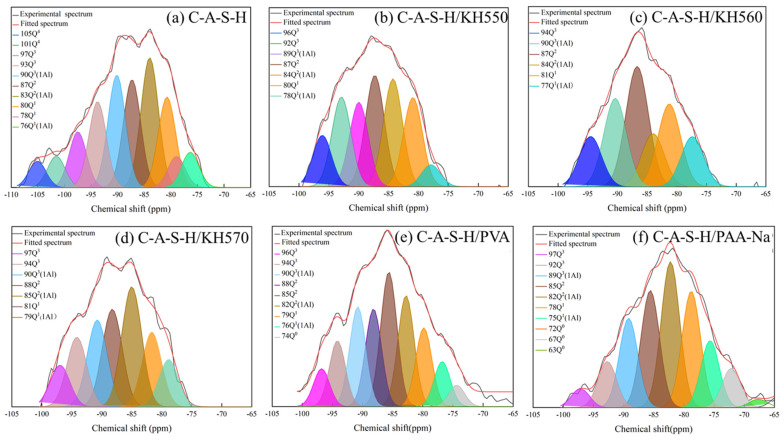
^29^Si NMR spectra for (**a**) C-A-S-H; (**b**) C-A-S-H/KH570; (**c**) C-A-S-H/KH570; (**d**) C-A-S-H/KH570; (**e**) C-A-S-H/PVA; (**f**) C-A-S-H/PAA-Na.

**Figure 19 polymers-14-04214-f019:**
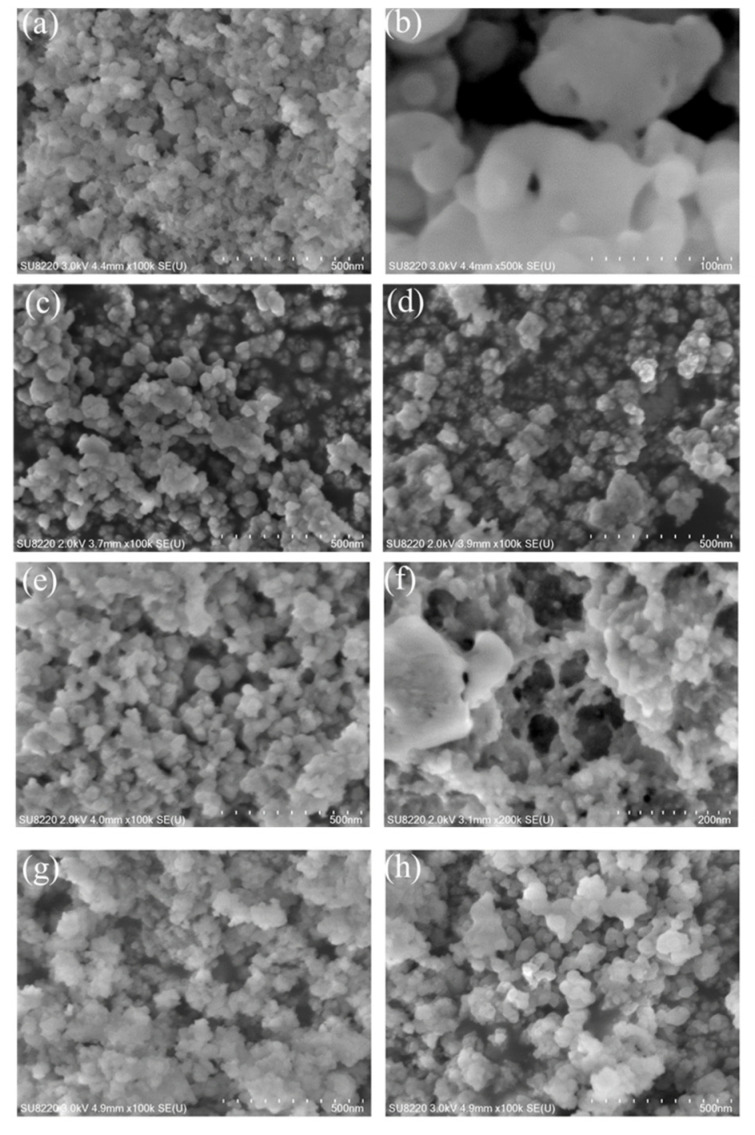
FE-SEM images (**a**,**b**) C-A-S-H; (**c**) C-A-S-H/KH550; (**d**) C-A-S-H/KH560 (**e**,**f**) C-A-S-H/KH570; (**g**) C-A-S-H/PVA; (**h**) C-A-S-H/PAA-Na.

**Table 1 polymers-14-04214-t001:** The chemical composition of slag.

Components (%)	CaO	SiO_2_	Al_2_O_3_	MgO	SO_3_	Fe_2_O_3_	TiO_2_	MnO	Others
Slag	37.12	32.06	14.84	10.77	1.67	1.14	0.92	0.33	1.15

**Table 2 polymers-14-04214-t002:** Mixing proportions of organic–inorganic composite geopolymer paste.

Samples	Slag	NaOH	Waterglass	BaCl_2_	Water	Polymer			
	(g)	(g)	(g)	(g)	(g)	(wt%)			
Reference1	4000	196.99	976.16	40	479.80	0			
Reference2	4000	178.65	1314.38	40	396.10	0			
Reference3	4000	161.60	1285.20	40	317.70	0			
Reference4	4000	134.75	1392.05	40	262.70	0			
Reference5	4000	53.87	428.56	40	772.50	0			
Reference6	4000	107.69	856.72	40	545.10	0			
Reference7	4000	161.60	1285.20	40	317.70	0			
Reference8	4000	215.37	1713.37	40	90.70	0			
PVA	4000	161.60	1285.20	40	317.70	1	3	5	7
PAA-Na	4000	161.60	1285.20	40	317.70				
KH550	4000	161.60	1285.20	40	317.70				
KH560	4000	161.60	1285.20	40	317.70				
KH570	4000	161.60	1285.20	40	317.70				

**Table 3 polymers-14-04214-t003:** The compressive strength and flexural strength of organic–inorganic composite geopolymer.

Samples	Polymer Dosage ^a^	Compressive Strength * (MPa)		Flexural Strength * (MPa)	
		7 d	28 d	7 d	28 d
Reference	0%	63.73 ± 1.5	70.34 ± 2.1	4.21 ± 0.2	5.39 ± 0.4
Slag/PVA	1%	80.24 ± 1.2	83.35 ± 1.5	4.38 ± 0.1	6.22 ± 0.2
Slag/PVA	3%	92.32 ± 2.2	97.38 ± 0.9	5.34 ± 0.2	7.80 ± 0.1
Slag/PVA	5%	96.97 ± 0.9	110.56 ± 1.2	6.34 ± 0.1	7.64 ± 0.2
Slag/PVA	7%	95.33 ± 1.4	97.23 ± 2.3	5.45 ± 0.4	6.76 ± 0.3
Slag/PAA-Na	1%	75.80 ± 1.3	80.40 ± 1.6	4.59 ± 0.3	6.40 ± 0.3
Slag/PAA-Na	3%	80.56 ± 1.9	82.34 ± 1.6	5.78 ± 0.1	7.23 ± 0.3
Slag/PAA-Na	5%	100.67 ± 1.1	109.34 ± 0.7	7.24 ± 0.2	8.31 ± 0.2
Slag/PAA-Na	7%	82.32 ± 2.1	82.46 ± 1.1	5.26 ± 0.3	7.20 ± 0.3
Slag/KH550	1%	90.56 ± 2.3	92.46 ± 2.1	5.34 ± 0.2	7.23 ± 0.5
Slag/KH550	3%	95.67 ± 2.4	100.67 ± 1.1	5.46 ± 0.4	8.70 ± 0.3
Slag/KH550	5%	116.57 ± 2.1	113.24 ± 1.3	6.81 ± 0.0	10.38 ± 0.1
Slag/KH550	7%	100.28 ± 2.3	100.67 ± 1.3	6.32 ± 0.5	8.80 ± 0.1
Slag/KH560	1%	66.68 ± 1.7	65.34 ± 1.3	4.22 ± 0.1	5.66 ± 0.1
Slag/KH560	3%	68.43 ± 3.1	72.34 ± 1.2	3.28 ± 0.1	5.32 ± 0.5
Slag/KH560	5%	76.16 ± 1.2	75.17 ± 3.0	4.82 ± 0.2	7.85 ± 0.3
Slag/KH560	7%	66.79 ± 1.4	70.35 ± 1.6	4.10 ± 0.1	6.34 ± 0.5
Slag/KH570	1%	98.34 ± 1.4	104.45 ± 1.1	7.56 ± 0.1	8.67 ± 0.1
Slag/KH570	3%	100.45 ± 1.5	110.34 ± 1.5	7.81 ± 0.0	9.46 ± 0.1
Slag/KH570	5%	124.99 ± 1.2	129.31 ± 2.7	8.56 ± 0.1	10.88 ± 0.0
Slag/KH570	7%	108.67 ± 0.9	112.37 ± 1.0	7.12 ± 0.2	9.78 ± 0.1

^a^ % of slag by mass; * Data presented are average ± standard deviation.

**Table 4 polymers-14-04214-t004:** The content of different kinds of SCAs and water-soluble polymers in solution by weight (%).

Polymer	The Supernatant	The Elution Solution		The Chemical Adsorption Amount
		25 °C Water	100 °C Water	
KH550	0	0.19	0.11	99.7
KH560	0	18.58	10.42	71
KH570	0	0	0	100
PVA	9.4	0	0	90.6
PAA-Na	5.87	0	0	94.13

**Table 5 polymers-14-04214-t005:** Oxide composition of each powder formulation as determined by X-ray fluorescence analysis. An error of approximately 1wt% is expected.

Specimen	The Actual Proportions				
	SiO_2_	CaO	Al_2_O_3_	C/S	A/S
C/S = 0.8 A/S = 0.2	42.08	28.48	4.83	0.701	0.135
C/S = 0.8 A/S = 0.2 KH550	43.78	26.96	5.28	0.660	0.142
C/S = 0.8 A/S = 0.2 KH560	45.35	24.47	5.32	0.608	0.138
C/S = 0.8 A/S = 0.2 KH570	44.20	28.67	5.41	0.695	0.144
C/S = 0.8 A/S = 0.2 PVA	42.63	29.90	5.07	0.751	0.140
C/S = 0.8 A/S = 0.2 PAA-Na	37.97	29.16	4.42	0.823	0.137

**Table 6 polymers-14-04214-t006:** ^29^Si MAS NMR parameters for C-A-S-H nanocomposites.

Sample	Q^n^ Cumulative Intensity							
	Q^1^	Q^1^ (1Al)	Q^2^	Q^2^(1Al)	Q^3^	Q^3^ (1Al)	Q^2^/Q^1 a^ MCL	Al/Si
C-A-S-H	16.69	4.83	14.83	17.83	19.47	15.47	1.51 6.98	0.17
C-A-S-H/KH550	15.81	3.88	19.73	19.11	25.01	14.96	1.97 8.12	0.15
C-A-S-H/KH560	18.59	8.88	27.03	11.93	11.30	19.87	1.31 6.83	0.20
C-A-S-H/KH570	10.29	5.05	17.91	16.91	16.33	17.98	2.37 10.41	0.21
C-A-S-H/PVA	11.13	6.38	32.88	15.7	14.63	14.09	2.77 12.14	0.14
C-A-S-H/PAA-Na	17.23	9.01	17.36	21.65	9.59	13.22	1.67 7.78	0.15

a: Mean chain length.

## Data Availability

Not applicable.

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
