# Peer review of "Effect of Organic Polymers on Mechanical Property and Toughening Mechanism of Slag Geopolymer Matrix"

_polymers, 2022, doi:10.3390/polym14194214_

Round 1

Reviewer 1 Report

The authors evaluate the possible use of two chemically reactive polymers to increase the properties of slag geopolymeric paste. The influence of polymers is investigated in terms of fluidity, rheological and mechanical properties and porosity. The characterisations of the different mixtures are very detailed. I would like to congratulate the authors on a meticulous and well-done paper.

I found some errors in the text:

·         Line 15-16: Correct the sentence

·         Line 168: What is “admixute”?

Here are a few suggestions:

·         Have you not thought about the environmental impact of these products?

·         Due to the different chemical nature of these products, has the durability of these newly proposed pastes not been evaluated? Why? Have you thought about it?

Regards

Author Response

Dear respected Editor and Reviewers:

Thank you very much for your comments on our paper and your kind consideration. We also appreciate the valuable comments of the reviewers. According to the comments, we modified the manuscript and marked the revised portions in red. Our response is presented as follows:

Reviewer #1: The authors evaluate the possible use of two chemically reactive polymers to increase the properties of slag geopolymer paste. The influence of polymers is investigated in terms of fluidity, rheological and mechanical properties and porosity. The characterizations of the different mixtures are very detailed. I would like to congratulate the authors on a meticulous and well-done paper.

Response: Thank you very much for the comments about the contents, all of which were taken into consideration in detail. According to the comments, our response on a point-by-point basis is as follows.

Point 1: Line 15-16: Correct the sentence

Response 1: Thanks for the comments. The relevant sentence was modified as follows: “The microscopic morphology of the geopolymer matrix was analyzed using Scanning Electron Microscopy (SEM).”

Point 2: Line 168: What is “admixture”?

Response 2: Thanks for the comments. In order to delay hydration, barium chloride (BaCl2) was added as a set retarder.

Point 3:  Have you not thought about the environmental impact of these products?

Response 3: Thanks for the comments. The authors are currently concentrating on the mechanism of interaction between the incorporation of organic materials with different functional groups and the slag-based polymers, and will subsequently focus on the durability and environmental impact of the products.

Point 4:  Due to the different chemical nature of these products, has the durability of these newly proposed pastes not been evaluated? Why? Have you thought about it?

Response 4: Thanks for the comments. The authors agree with the reviewers’ comments. The authors are currently concentrating on experiments for durability against sulfate attack, carbonation, wet-dry cycling condition, etc.

Reviewer 2 Report

The paper is entitled “Effect of Organic Polymers on Mechanical Property and Toughening Mechanism of Slag Geopolymer Matrix” so it describes sufficiently the content and what the reader may expect. The only confusion is the term “mechanical property” (used in singular in the title and in the conclusions chapter), while in the abstract there are mentioned “mechanical properties” (in plural), and in fact the Authors did research only on flexural and compressive strength. In fact, the phrase “mechanical properties” is much more than only these two strengths.

In general, the paper is mainly written well, but in some cases, the sentences are too long (multiple-complex sentences) making them incomprehensible to the reader.

My comments (the order is chronological in relation to the text of the paper):

1.     The content of the Figure 1 is not the original achievement of the Authors and the title “Proposed chemical reactions of …” indicates it.

2.     Line 138: the modulus of water glass was 3.2. Why do Authors decide for such a value of the modulus? (according to the literature, the lower modulus may refer to higher strength)

3.     Line 144: What is the explanation for assuming these values of concentrations? 

4.     Table 1: The percentage of the components is lower than 100%, so probably the value “Others” should be increased.

5.     Line 154 (and subsequently in the whole body of the paper): what do Authors mean by “ambient temperature”?

6.     Line 155: reference [26] – why it refers to the paper describing standard test method for compressive strength of cement mortars and not directly to the standard itself?

7.     Line 157: the mass ratio of water to slag is 0.25. Is there any justification for such a ratio?

8.     Line 176: why the specimens were cured by steam curing? Usually the geopolymers are cured in classic laboratory ovens (the temperature has significant meaning). So, in this case, what was the moisture content and has it any significant meaning?

9.     Line 178: the dimensions of the samples for testing the compressive and flexural strength are surprising. The typical dimensions for testing compressive and flexural strength of cement and non-cementitious materials (mortars) are: 160mm x 40mm x 40mm. Especially the square cross-section (40mm by 40mm) is required for compressive strength testing. Using the samples of different than standard dimensions, leads to difficulties with comparing the results with other researchers (it is hard to implement the scale effect).

10.  Figure 3: what about NaOH?

11.  Line 216: the Chinese Standard GB/T 8077-2012 should be also listed in the references.

12.  Line 224-225: the loading speed should be expressed in N/s and not in mm/min.

13.  Line 233: the frequency unit should be MHz (not MHZ).

14.  Line 286: the dimensions of the samples for testing the compressive and flexural strength – please see the comment no. 9 (also, the dimensions are different than the ones listed in line 178).

15.  The descriptions in Figure 9, Figure 10, Figure 14 to Figure 19 seems to be too small.

16.  Line 627: the flexural strength of 10.85 – there is no such a value in the Table 3.

17.  And the final (general) remark. The paper reminds rather well-presented research report than the scientific paper. Moreover, the originality and novelty of research activity is not well exposed and/or highlighted. Presented conclusions are related to the performed laboratory tests - which is good of course. However, in the scientific paper there are also expected some general conclusions, indicating the contribution to the development of the relevant field of science. Otherwise, the results of the investigations might be limited only to the local application, while the research described in the paper would be of much greater practical importance.

Author Response

Dear respected Editor and Reviewers:

Thank you very much for your comments on our paper and your kind consideration. We also appreciate the valuable comments of the reviewers. According to the comments, we modified the manuscript and marked the revised portions in red. Our response is presented as follows:

Reviewer #2: The paper is entitled “Effect of Organic Polymers on Mechanical Property and Toughening Mechanism of Slag Geopolymer Matrix” so it describes sufficiently the content and what the reader may expect. The only confusion is the term “mechanical property” (used in singular in the title and in the conclusions chapter), while in the abstract there are mentioned “mechanical properties” (in plural), and in fact the Authors did research only on flexural and compressive strength. In fact, the phrase “mechanical properties” is much more than only these two strengths.

In general, the paper is mainly written well, but in some cases, the sentences are too long (multiple-complex sentences) making them incomprehensible to the reader.

My comments (the order is chronological in relation to the text of the paper):

Response: Thanks for the comments. The authors apologize for the mistake. The mechanical property in this paper is only the compressive and flexural strength, other mechanical properties will continue to be studied in the future. “Mechanical properties” was replaced with “mechanical property” in the paper.

Point 1: The content of the Figure 1 is not the original achievement of the Authors and the title “Proposed chemical reactions of …” indicates it.

Response 1: Thanks for the comments. The authors deleted Figure 1 due to incorrect citation.

Point 2: Line 138: the modulus of water glass was 3.2. Why do Authors decide for such a value of the modulus? (according to the literature, the lower modulus may refer to higher strength)

Response 2: Thanks for the comments. The initial modulus of the purchased water glass was 3.2, and the author added sodium hydroxide (NaOH) to adjust the modulus to 1.2-2.0 to investigate the effect of modulus on strength.

Point 3: Line 144: What is the explanation for assuming these values of concentrations?

Response 3: Thanks for the comments. The relevant expressions were added to the revised manuscript. In order to obtain C-A-S-H gel with C/S=0.8 and A/S=0.2 which is consistent with the composition of the C-A-S-H gels produced by the alkali activation reaction of the real slag-based geopolymer.

Point 4: Table 1: The percentage of the components is lower than 100%, so probably the value “Others” should be increased.

Response 4: Thanks for the comments. The authors apologize for the mistake in Table 1. The relevant expressions were corrected in the revised manuscript.

Point 5: Line 154 (and subsequently in the whole body of the paper): what do Authors mean by “ambient temperature”?

Response 5: Thanks for the comments. The ambient temperature means 25 oC. The relevant expressions were added to the revised manuscript.

Point 6: Line 155: reference [26] – why it refers to the paper describing standard test method for compressive strength of cement mortars and not directly to the standard itself ?

Response 6: Thanks for the comments. The authors agree with the reviewers’ comments. Reference [26] was replaced with: ASTM C109/C109M-16a. Standard Test Method for Compressive Strength of Hydraulic Cement Mortars (Using 2-in. Or [50-mm] Cube Specimens). ASTM International, West Conshohocken, PA. 2016. https://doi.org/10.1520/C0109_C0109M16A.

Point 7: Line 157: the mass ratio of water to slag is 0.25. Is there any justification for such a ratio?

Response 7: Thanks for the comments. The samples were created using the vibration pressure machine. Based on the authors’ research, the samples cannot be stirred at lower water-to-slag ratios and higher pressure would lead to leakage of water at higher water-to-slag ratios.

 Point 8: Line 176: why the specimens were cured by steam curing? Usually, the geopolymers are cured in classic laboratory ovens (the temperature has significant meaning). So, in this case, what was the moisture content and has it any significant meaning?

Response 8: Thanks for the comments. Slag-based polymers are prone to dry shrinkage, so the authors utilized steam curing at 80 oC to reduce shrinkage according to the reference [27-28]. In addition, the moisture content was 100%.

[27] A. Marcialis; L. Massidda; U.Sanna. Low temperature steam curing hydration of lime-fly-ash compacts. Cem. Concr. Res. 1983; 13, 887-894. https://doi.org/10.1016/0008-8846(83)90090-X

[28] A. Ferhat Bingöl; Ä°lhan Tohumcu. Effects of different curing regimes on the compressive strength properties of self compacting concrete incorporating fly ash and silica fume. Mater. Design. 2013, 51, 12-18. https://doi.org/10.1016/j.matdes.2013.03.106.

 Point 9: Line 178: the dimensions of the samples for testing the compressive and flexural strength are surprising. The typical dimensions for testing compressive and flexural strength of cement and non-cementitious materials (mortars) are: 160mm x 40mm x 40mm. Especially the square cross-section (40mm by 40mm) is required for compressive strength testing. Using the samples of different than standard dimensions, leads to difficulties with comparing the results with other researchers (it is hard to implement the scale effect).

Response 9: Thanks for the comments. Geopolymers do not have a test standard, so we can only compare the data longitudinally to determine the overall trend. Additionally, the samples were prepared by the vibration pressure machine to remove macroscopic pores and promote organic-inorganic cross-linking interactions. The size of the mold was fixed in order to accommodate the vibration space.

Point 10: Figure 3: what about NaOH?

Response 10: Thanks for the comments. The authors did not depict the additional sodium hydroxide (NaOH) because the water glass in Figure 3 is one with a well-adjusted modulus using NaOH.

Point 11: Line 216: the Chinese Standard GB/T 8077-2012 should be also listed in the references.

Response 11: Thanks for the comments. The reference listed further as follows:

[28]GB/T 8077-2012 Domestic - National Standards - State Administration of Market Supervision and Administration CN-GB.

Point 12: Line 224-225: the loading speed should be expressed in N/s and not in mm/min.

Response 12: Thanks for the comments. The loading speed was displacement controlled, so the units were mm/min.

Point 13: Line 233: the frequency unit should be MHz (not MHZ).

Response 13: Thanks for the comments. The authors apologized for the mistake. The relevant expressions were modified to the revised manuscript.

Point 14: Line 286: the dimensions of the samples for testing the compressive and flexural strength – please see the comment no. 9 (also, the dimensions are different than the ones listed in line 178).

Response 14: Thanks for the comments. The authors apologized for the mistake. The dimensions of the samples for testing the compressive and flexural strength were 25 mm ´ 25 mm´ 25 mm and 250 mm ´ 50 mm ´ 25 mm. The relevant expressions were changed to the revised manuscript.

Point 15: The descriptions in Figure 9, Figure 10, Figure 14 to Figure 19 seems to be too small.

Response 15: Thanks for the comments. The related figures were modified in the revised manuscript.

Point 16: Line 627: the flexural strength of 10.85 – there is no such a value in the Table 3.

Response 16: Thanks for the comments. The authors apologize for the mistake. The related text was modified in the revised manuscript as follows: “When the polymer was 5 wt%, the optimal mechanical properties appeared at the slag/KH570 paste with compressive strength of 129.31 MPa and the flexural strength of 10.88 MPa at 28 d.”

Point 17: And the final (general) remark. The paper reminds rather well-presented research report than the scientific paper. Moreover, the originality and novelty of research activity is not well exposed and/or highlighted. Presented conclusions are related to the performed laboratory tests - which is good of course. However, in the scientific paper there are also expected some general conclusions, indicating the contribution to the development of the relevant field of science. Otherwise, the results of the investigations might be limited only to the local application, while the research described in the paper would be of much greater practical importance.

Response 17: Thanks for the comments. The authors agree with the reviewers’ comments. It was very helpful for us to clarify the research significance of designing organic-inorganic composite gelling materials. The related text was modified in the revised manuscript as follows: “The research focuses on the qualitative and quantitative characterization of the chemical reaction processes between organics with different functional groups and geopolymers in order to establish a theoretical foundation for subsequent organic-toughening geopolymers. In the series of silane coupling agents (SCAs), the “X” group and the “RO” groups of silanes with a double bond and “-CH3” were selected to bridge the inorganic components which can promote the toughness of slag-based geopolymer. Additionally, in the series of water-soluble polymers, the “COO-” group was selected to improve the toughness of slag-based geopolymer.”

Round 2

Reviewer 2 Report

I would like to thank Authors for their replies and clarifications. I accept their response to my comments, except Response 9 (the dimensions of the samples for strength tests). That is true, that there is no standard for testing flexural strength nor compressive strength for geopolymers. However, it is commonly assumed that the standard for testing cement mortars is applied for this testing. I do not know what “overall trend” are the Authors writing about. To test the strength properties on the samples with dimensions different than most of other researches do, is completely useless – because of the scale effect there is no way to compare the results of the testing. It is not an explanation that dimensions of the samples were accommodated to the vibration space – this only means that the testing was poorly planned.

Author Response

Response to Reviewer 2 Comments

Dear respected Editor and Reviewers:

Thank you very much for your comments on our paper and your kind consideration. We also appreciate the valuable comments of the reviewers. According to the comments, we modified the manuscript and marked the revised portions in red. Our response is presented as follows:

Reviewer #2: I would like to thank Authors for their replies and clarifications. I accept their response to my comments, except Response 9 (the dimensions of the samples for strength tests). That is true, that there is no standard for testing flexural strength nor compressive strength for geopolymers. However, it is commonly assumed that the standard for testing cement mortars is applied for this testing. I do not know what “overall trend” are the Authors writing about. To test the strength properties on the samples with dimensions different than most of other researches do, is completely useless – because of the scale effect there is no way to compare the results of the testing. It is not an explanation that dimensions of the samples were accommodated to the vibration space – this only means that the testing was poorly planned.

 Point 9: Line 178: the dimensions of the samples for testing the compressive and flexural strength are surprising. The typical dimensions for testing compressive and flexural strength of cement and non-cementitious materials (mortars) are: 160mm x 40mm x 40mm. Especially the square cross-section (40mm by 40mm) is required for compressive strength testing. Using the samples of different than standard dimensions, leads to difficulties with comparing the results with other researchers (it is hard to implement the scale effect).

Response: Thank you so much for the comments. Although the standards for cement are typically borrowed, there is no universal standard for geopolymer-based organic-inorganic composite cementitious materials, and the dimensions frequently differ from those measured for cement due to variations in production methods, such as the hot press molding process [1]. The size of the sample prepared by Hao Pan was 160*40*7mm [2].

The authors prepared the samples using the vibration pressure machine to remove macroscopic pores and promote organic-inorganic cross-linking interactions. The length, width and height of the internal machine tooling for this molding process are 300*300*30 mm, so the strength cannot be tested to the standard size at present. Additionally, the pores cannot be fully discharged during pressing for samples with excessive thickness. Samples were cut to the same size for longitudinal comparison of the effect of the organic addition on the compressive and flexural strength. The authors will further optimize the preparation process and improve the condition of the equipment later.

[1] O. Ekinciogl; M. H. Ozkul; L. J. Struble; S. Patachia. State of the art of macro-defect-free composites. J. Mater. Sci. 2018,53,10595-10616. https://doi.org/10.1007/s10853-018-2328-y

[2] H. Pan; W. She; W. Zuo; Y. Zhou; J. Liu. Hierarchical Toughening of a Biomimetic Bulk Cement Composite. ACS Appl. Mater. 2020, 12, 53297-53309. https://dx.doi.org/10.1021/acsami.0c15313
